# Prognostic Significance of *GPR55* mRNA Expression in Colon Cancer

**DOI:** 10.3390/ijms23094556

**Published:** 2022-04-20

**Authors:** Hager Tarek H. Ismail, Manar AbdelMageed, Gudrun Lindmark, Marie-Louise Hammarström, Sten Hammarström, Basel Sitohy

**Affiliations:** 1Department of Clinical Microbiology, Infection and Immunology, Umeå University, SE-90185 Umeå, Sweden; hager_vet@hotmail.com (H.T.H.I.); mamohammad@zu.edu.eg (M.A.); marie-louise.hammarstrom@umu.se (M.-L.H.); sten.hammarstrom@umu.se (S.H.); 2Department of Radiation Sciences, Oncology, Umeå University, SE-90185 Umeå, Sweden; 3Department of Clinical Pathology, Faculty of Veterinary Medicine, Zagazig University, Zagazig 44511, Egypt; 4Department of Pathology, Faculty of Veterinary Medicine, Zagazig University, Zagazig 44511, Egypt; 5Institution of Clinical Sciences, Lund University, SE-25187 Helsingborg, Sweden; lindmarkgudrun@gmail.com

**Keywords:** colon cancer, GPR55, CEA, CXCL17, CXCL16, qRT-PCR, regional lymph nodes, prognosis

## Abstract

G protein-coupled receptor 55 (GPR55) probably plays a role in innate immunity and tumor immunosurveillance through its effect on immune cells, such as T cells and NK cells. In this study, the prognostic value of GPR55 in colon cancer (CC) was investigated. mRNA expression levels of GPR55 were determined in 382 regional lymph nodes of 121 CC patients with 12 years observation time after curative surgery. The same clinical material had previously been analyzed for expression levels of CEA, CXCL16, CXCL17, GPR35 V2/3 and LGR5 mRNAs. Clinical cutoffs of 0.1365 copies/18S rRNA unit for GPR55 and 0.1481 for the GPR55/CEA ratio were applied to differentiate between the high- and low-GPR55 expression groups. Kaplan–Meier survival analysis and Cox regression risk analysis were used to determine prognostic value. Improved discrimination between the two groups was achieved by combining GPR55 with CEA, CXCL16 or CXCL17 compared with GPR55 alone. The best result was obtained using the GPR55/CEA ratio, with an increased mean survival time of 14 and 33 months at 5 and 12 years observation time, respectively (*p* = 0.0003 and *p* = 0.003) for the high-GPR55/CEA group. The explanation for the observed improvement is most likely that GPR55 is a marker for T cells and B cells in lymph nodes, whereas CEA, CXCL16 and CXCL17, are markers for tumor cells of epithelial origin.

## 1. Introduction

G protein-coupled receptor 55 (GPR55) is a member of a large family of signaling receptors in humans, of which several members are targets for approved drugs [1]. GPR55 is structurally related to GPR35, GPR92 and GPR23 [2]. Recent single-cell sequencing studies have revealed that in colon, GPR55 is expressed predominantly in T cells and, to some extent, in goblet cells, distal enterocytes and enteroendocrine cells [3]. In lymph nodes, only T cells and B cells express GPR55 [3], which is in agreement with earlier studies showing that GPR55 is expressed in immune cells, including plasma cells and NK cells [3,4,5].

GPR55 acts as regulator in innate immunity and tumor immunosurveillance via stimulatory effects in immune cells, such as T cells and NK cells [3,4,5,6]. In mice, GPR55 was found to be abundantly expressed by all types of small intestine intraepithelial lymphocytes (IELs), with the highest expression in CD8γδT-cells [6]. Here, GPR55 mediates migration inhibition of IELs in response to the natural ligand lysophosphatidylinositol (LPI) and negatively regulates TCRγδ IEL accumulation in the small intestine [6]. Because human γδT cells also express GPR55, it is possible that GPR55 has similar functions in humans [3,5]. LPI was reported as the endogenous agonist of GPR55, being expressed in various cancers in an aggressiveness-related manner, suggesting a role of the LPI/GPR55 axis in cancer [7]. Whether the level of *GPR55* mRNA in the primary tumor of colorectal cancer (CRC) patients is associated with positive or negative prognosis has been investigated in three studies [3,8,9]. In one study, the authors found a significant association between high GPR55 mRNA levels and negative prognosis [8]; in the other two studies, the authors found no significant difference.

In this study we reinvestigated the relationship between colon cancer (CC) and GPR55, focusing on analysis of patients’ lymph nodes obtained from curative surgery. We developed a new highly specific quantitative reverse transcriptase–polymerase chain reaction (qRT-PCR) assay for GPR55 with an RNA copy standard to determine absolute expression levels using the amounts of the housekeeping gene 18S rRNA for normalization. Low levels of GPR55 mRNA were found to be associated with negative prognosis. Furthermore, a combination of GPR55 mRNA analysis and analysis of mRNA for other biomarkers, such as CXCL16, CXCL17 and CEA, significantly improved the prognostic value.

## 2. Results

### 2.1. GPR55 mRNA Expression Levels in Primary Colon Cancer Tumors and Colon Cancer Cell Lines

The median mRNA expression level of GPR55 was slightly higher in primary CC tumors than in normal colon tissues, but the difference was not statistically significant (0.03 and 0.01 mRNA copies/18S rRNA unit, respectively; *p* = 0.2) (Figure 1A). GPR55 mRNA was absent in 4/6 CC cell lines (LS174T, Caco2, DLD1 and HCT8) and barely detected in two CC cell lines (HT29 and T84). Similarly, a foreskin fibroblast cell line (FSU), an endothelial cell line (HUVEC) and a monocyte cell line (U937) lacked GPR55 expression. In contrast, GPR55 mRNA was expressed at high levels in both B cell lines (CNB6 and B6) and in a T cell line (Jurkat). These results suggest that GPR55 mRNA expression in primary CC tumors and normal colon tissue is, to a large extent, due to the presence of immune cells in these tissues. Whether the low levels of GPR55 found in CC cell lines HT29 and T84 are derived from goblet-cell-like tumor cells is an open question.

### 2.2. GPR55 mRNA Expression Levels in Regional Lymph Nodes of Colon Cancer Patients

The mRNA expression level of *GPR55* was determined in 382 regional lymph nodes from 121 CC patients representing all four TNM stages and in 77 lymph nodes from 13 patients with non-cancerous intestinal disease. Median expression levels of GPR55 mRNA were 0.41, 0.35, 0.36 and 0.25 mRNA copies/18S rRNA unit in TNM stages I, II, III and IV, respectively. Lymph nodes of control patients expressed 0.34 mRNA copies/18S rRNA unit. There was no significant difference in expression levels between any of the four TNM stages and control lymph nodes (Figure 1B). The 382 lymph nodes were then stratified by histopathology into H&E(+) lymph nodes, where cancer cells were observed (*n* = 22), and H&E(−) lymph nodes, where no cancer cells were detected (*n* = 360). Figure 1C shows that the median GPR55 mRNA expression level was ≈twice as high in H&E(−) lymph nodes compared to H&E(+) lymph nodes (0.37 and 0.21 mRNA copies/18S rRNA unit, respectively, *p* = 0.0002). No significant difference was found between the expression levels of GPR55 mRNA based on gender either in the lymph nodes or the primary tumors (*p* > 0.05).

To further investigate whether GPR55 mRNA expression was correlated to the presence of tumor cells or immune cells in lymph nodes from CC patients, the nodes were divided into three groups based on their previously determined CEA mRNA expression levels: CEA(+), CEA(int) and CEA(−). CEA mRNA levels in the first group were above the clinical cutoff (>3.67 CEA mRNA copies/18S rRNA unit) and between 0.013–3.67 and <0.013 mRNA copies/18S rRNA unit in the second and third group, respectively [10,11]. GPR55 mRNA levels were significantly decreased in both the CEA(+) group and the CEA(int) group compared to the CEA(−) group (*p* < 0.0001). The median levels of *GPR55* mRNA were 0.14, 0.35 and 0.37 mRNA copies/18S rRNA units, respectively (Figure 1D).

Because GPR55 appears to be predominantly expressed in immune cells and not in CC tumor cells, whereas CEA is expressed in CC tumor cells but not in immune cells [7,10,11], we investigated whether calculating the GPR55 mRNA/CEA mRNA ratios would sharpen the difference between TNM stages of CC patients. Figure 2A shows the result. Each patient is represented by the lymph node with the highest expression level of CEA mRNA (121 nodes from 121 patients). Lymph nodes of control patients were dealt with in the same way. Stage IV patients had significantly lower ratios compared to patients in stage I (*p* = 0.001), stage II (*p* = 0.004) and stage III (*p* = 0.03). No significant difference was found between patients in stage III, stage I and stage II, although there was a trend toward lower values in stage III. The median ratios were 24.88, 16.68, 7.39 and 0.001 mRNA copies/18S rRNA unit in TNM stages I, II, III and IV, respectively (Figure 2A). The median ratio of GPR55/CEA mRNA was 3 × 10^4^ times higher in the H&E(−) group compared to the H&E(+) lymph nodes group (ratios 94.07 and 0.0032, respectively, *p* < 0.0001) (Figure 2B). The results indicate that calculating a ratio of this nature can help to reveal heterogeneities within TNM stages, which could be highly relevant for prognosis.

### 2.3. Correlation between mRNA Expression Levels of GPR55 and Each of CEA, CXCL17, CXCL16, GPR35 V2/3 and LGR5 in Regional Lymph Nodes of Colon Cancer Patients

The mRNA expression levels of CEA, CXCL17, CXCL16, GPR35 V2/3 and LGR5 have been previously determined in the same 382 lymph nodes studied in this work [10,12,13,14]. GPR55 mRNA showed a significant negative correlation with the other biomarker mRNAs, except for LGR5 mRNA, in stage IV lymph nodes, a finding which is expected because GPR55 is not expressed or expressed at very low levels in CC tumor cells (see above). Conversely, GPR55 was positively correlated with these biomarker mRNAs in lymph nodes of stage I patients known to generally lack tumor cells (Table 1). Principal component analysis (PCA) with varimax rotation was performed with the four variables, GPR55, CXCL16, CXCL17 and CEA. The results extracted two principal components; the first showed 31% of variance in data, and the second showed 25% variance. GPR55, CXCL16 and CXCL17 mRNA variables were related to the first component and showed that CXCL16 and CXCL17 were negatively related to GPR55 (−0.52). CEA mRNA was related to the second component and showed a negative relation with GPR55 (−0.48) (Appendix A).

### 2.4. Clinical Significance of GPR55 mRNA Expression Levels and of GPR55/CEA mRNA Expression Ratios in Lymph Nodes for Colon Cancer Patients: Prediction of Recurrence

To determine whether GPR55 mRNA expression levels in regional lymph nodes of CC patients could be used to predict disease recurrence and survival time after surgery, the hazard risk ratio was calculated using Cox regression analysis. Additionally, the data were subjected to analysis by the Kaplan–Meier survival model combined with the log-rank test. Each patient was represented by the lymph node with the lowest expression level of GPR55 mRNA, and a cutoff level discriminating between patients with high and low risk for recurrence was empirically determined. Similarly, the GPR55 mRNA/CEA mRNA ratio was calculated, and patients were allocated into GPR55(+) and GPR55(−) groups. Summaries of these survival analyses are shown in Table 2 and Appendix A.

Figure 3A shows Kaplan–Meier analysis of all 121 CC patients divided into a GPR55(−) and a GPR55(+) group using the 28th percentile of the values for division (=0.1365 GPR55 mRNA copies/18S rRNA unit). Patients in the high-expression group (GPR55(+) group, *n* = 88) showed a 0.6-fold decreased recurrence rate compared to the low-expression group (GPR55(−) group, *n* = 33) when followed for five years and 0.5-fold at a follow-up time of 12 years (*p* = 0.1 and *p* = 0.05, respectively). Only at 12 years was the mean survival time difference between the two groups significant (i.e., 14 months; *p* = 0.04) (Figure 3A).

However, if the analysis was confined to the group of CC patients that expressed high levels of CXCL16 mRNA (i.e., CXCL16 mRNA values >7.2 copies/18S rRNA unit; [14]) it was found that the patients in the GPR55(+) group (*n* = 33) had 0.3- and 0.4-fold decreased recurrence rates compared to the low-expression group (GPR55(−) group, *n* = 15) when followed for 5 and 12 years after surgery, respectively (*p* = 0.02 and *p* = 0.04). Kaplan–Meier analysis revealed that mean survival times differed, with 16 and 40 months between the two groups (*p* = 0.01 and *p* = 0.04, respectively) (Figure 3B). It is interesting to note that combined analysis for these two markers actually identifies almost all patients that died from CC within the first 3 years in the CXCL16(+)/GPR55(−) group.

A similar result was obtained when analysis was performed on the subgroup of CC patients expressing high levels of CXCL17 mRNA (CXCL17 mRNA values higher than 0.0014 mRNA copies/18S rRNA unit; 75th percentile; [14]), again dividing the lymph node values into a GPR55(+) and a GPR55(−) group using the same division as above (Figure 3C). The recurrence rate was 0.3-fold lower in the GPR55(+) group (*n* = 16) when followed for 5 years and 12 years (*p* = 0.07 and *p* = 0.07, respectively), with increased mean survival time of 17 months after 5 years and 44 months after 12 years (*p* = 0.05) when compared to the GPR55(−) group (*n* = 16) (Figure 3C).

Other restrictions of the patient group, such as analyzing a CEA(−) subgroup (Table 2), a CEA(+) subgroup, a CXCL16(−) subgroup, a CXCL17(−) subgroup, a GPR35 V2/3(−) subgroup or an LGR5(−) subgroup (Appendix A), did not yield significant discrimination between GPR55(+) and GPR55(−) groups.

An alternative way of utilizing biomarker mRNA data for prognosis is to calculate a ratio between two markers and then divide the values into a positive and a negative group. Figure 4A shows an example in which the GPR55 value has been divided by the CEA value for the particular lymph node using the lymph node with the lowest GPR55 value for each patient. The ratios are then divided into a high-GPR55 group and a low-GPR55 group, using a value of 0.1481 GPR55 mRNA copies/18S rRNA unit for division. Patients in the GPR55(++) group (*n* = 102) showed a 0.3-fold decreased recurrence rate compared to the low-expression group (GPR55(−−) group, *n* = 19) when followed for five years and 0.4-fold at a follow-up time of 12 years (*p* = 0.001 and *p* = 0.004, respectively). A difference in mean survival times amounting to 14 months in 5 years and 33 months in 12 years after surgery (*p* = 0.0003 and *p* = 0.003, respectively) was observed based on Kaplan–Meier survival analysis (Figure 4A).

Figure 4B shows the result of a ratio calculation applied to the CXCL16(+) group, resulting in a 0.4-fold lower recurrence rate in the GPR55(++) group (*n* = 32) when followed for 5 and 12 years, respectively (*p* = 0.02 and *p* = 0.04). This was associated with 13- and 40-month increased survival times in 5 and 12 years, respectively (*p* = 0.02, *p* = 0.04) when compared to the GPR55(−−) group (*n* = 16).

Finally, ratio analysis was applied to a group of CC patients expressing low CXCL17 mRNA levels (Figure 4C). The difference in recurrence rate decreased to 0.2-fold in the high-expression group (GPR55(++) group, *n* = 85) compared to the low-expression group (GPR55(−−) group, *n* = 6) when patients were followed for five years, versus 0.3-fold when followed for 12 years (*p* = 0.004 and *p* = 0.05, respectively). The GPR55(++) group had increased mean survival times of 19 and 41 months (*p* = 0.001 and *p* = 0.04, respectively).

### 2.5. Absence of Association between Risk of Recurrence or Survival Time after Surgery and Levels GPR55 mRNA Expression in Primary Colon Cancer Tumors

No difference in recurrence risk or survival time was observed in CC patients divided into GPR55(+) and GPR55(−) patients using the median mRNA of primary CC tumors as cutoff (0.03 mRNA copies/18S rRNA unit) (data not shown).

## 3. Discussion

The result of this study is unexpected. We found that low levels of GPR55 mRNA was associated with poor prognosis. The finding contrasts with the results for all other biomarker mRNAs that we have studied so far, i.e., CEA, CXL17, CXCL16, LGR5, LGR4, GPR35V2/3 and EpCAM [10,12,13,14,15,16,17], for which high levels were associated with poor prognosis. Notably, this reverse correlation was only found in lymph nodes. In primary tumors, there was no significant difference between the high- and low-GPR55 groups, confirming the results reported by other investigators [3,18]. The explanation for the result in lymph nodes is probably quite simple. GPR55 is highly expressed in T cells and B cells, cell types that dominate in a healthy lymph node, whereas lymph nodes heavily infested with tumor cells contain relatively fewer lymphocytes. CC tumor cells contribute only marginally or not at all to GPR55 mRNA, depending on which subtype of tumor cells constitute the disseminated tumor cells in the node. Thus, more tumor cells in the node means fewer lymphocytes and therefore lower GPR55 levels and vice versa.

The finding that GPR55 is essentially a lymphocyte marker in lymph nodes can be used to complement biomarkers that identify CC tumor cells either by performing combined marker analysis or by calculating a marker ratio and using this ratio to allocate the patients into a positive or negative group. Figure 3 and Figure 4 show examples demonstrating that improved discrimination between patients can be achieved by using two biomarkers instead of one (compare Figure 3A and Figure 4A). The most promising result so far was achieved using the GPR55 mRNA/CEA mRNA ratio. Analyses of the entire CC population demonstrated that an increased ratio was associated with an increased survival of 14 and 33 months in the high-expression group (GPR55++) compared to the low expression group (GPR55(−−)) when followed for 5 and 12 years, respectively (Figure 4B). The significance of these differences was very strong. The prognostic value of GPR55 mRNA was also significantly increased when combined with measurement of CXCL17 and CXCL16 mRNA levels, particularly at the 5-year follow-up, with 13–19-month mean survival difference. These results provide information about patients who can be spared unnecessary adjuvant chemotherapy and its side effects and other patients who would benefit from adjuvant chemotherapy to avoid relapse.

Whether GPR55 has a physiological role in promoting positive prognosis or whether the positive effect is due to the fact that the biomarker reflects the absence of tumor cells is unclear. If GPR55 has a physiological effect, it is probably through its role in innate immunity and tumor immunosurveillance [5,6,19]. This study did not allow us to determine the mechanism(s) of GPR55 for positive prognosis. However, it is noteworthy that GPR55 was observed to have antiproliferative and proapoptotic effects on cholangiocarcinoma cells [20]. Further studies are needed to elucidate its role in CC cancer progression.

In conclusion, this study demonstrates that GPR55 may serve as a marker for positive prognosis, either alone or combined with CEA, CXCL16 and CXCL17.

## 4. Materials and Methods

### 4.1. Patients and Tissue Specimens for mRNA Analysis

Primary tumor specimens were collected from 70 CC patients (31 men and 39 women; median age, 74 years; range, 42–88 years) after surgery. None of the patients received chemotherapy or radiotherapy before surgery. Fourteen patients were in stage I (T1-2N0M0), 31 in stage II (T3-4N0M0), 19 in stage III (anyTN1-2M0) and 6 in stage IV (anyTanyNM1). The tumor samples, approximately 0.5 × 0.5 × 0.5 cm in size, were gathered instantly after resection, snap-frozen and kept at −70 °C until RNA extraction. Normal colon samples were from the proximal or distal resection margin of CC tumors of 30 patients (17 men and 13 women; median age, 72 years; range, 57–85 years) and treated the same way.

Lymph nodes were collected from the 121 CC patients (55 men and 66 women; median age, 73 years; range, 42–89 years). A total of 73 lymph nodes were from 23 patients in stage I, 190 nodes were from 52 patients in stage II, 88 nodes were from 37 patients in stage III, and 31 nodes were from 9 patients in stage IV. A total of 22 lymph nodes were judged positive for disseminated tumor cells by routine histopathology H&E(+), and 360 lymph nodes were judged as H&E(−). Control lymph nodes (*n* = 77) were from 13 patients (10 men and 3 women; median age, 23 years; range, 9–32 years). Eleven of the controls had ulcerative colitis, one had Crohn’s disease and one patient had lipoma.

### 4.2. Cell Lines

Five human CC cell lines (HCT8, HT29, LS174T, DLD1, Caco2 and T84); a T cell line, Jurkat; two B cell lines, CNB6 and KR4; a monocyte cell line, U937; primary foreskin fibroblasts (FSU); and an endothelial cell line, HUVEC, were cultured and analyzed for mRNA expression. Different culture conditions and sources are as described earlier [11,21,22].

### 4.3. Real-Time qRT-PCR

For absolute quantification of GPR55 mRNA in lymph nodes, we established real-time qRT-PCR assay using specific primers placed in different exons and a reporter dye-labeled probe hybridizing over the exon boundary in the amplicon and specific RNA copy standards for the quantification. The primers and probe sequences for GPR55 mRNA (NM_005683.3) were forward primer 5′-GCACCGGACCACCAACA-3′, reverse primer 5′-ACCTCCCCAGCATCACACA-3′ and probe 5′-TTCAATGGGCATGAATT-3′. The reporter dye was FAM, and the quencher dye was MGB. The size of the amplicon was 68 bp. The qRT-PCR profile was 60 °C for 5 min and 95 °C for 1 min, followed by 45 cycles of 95 °C for 15 s and 60 °C for 1 min. RNA oligonucleotides with sequences identical to those in the areas amplified in the qRT-PCR assays were custom-synthesized at Dharmacon (Lafayette, CO, USA). Serial dilutions of the RNA copy standards at concentrations from 10^2^ to 10^7^ copies per µL were included in each qRT-PCR run. Concentrations in unknown samples were set from the standard curve and expressed as copies of mRNA per µL. The concentration of 18S rRNA was expressed as arbitrary units from a standard curve of serial dilutions of a preparation of total RNA from human peripheral blood mononuclear cells. One unit was defined as the amount of 18S rRNA in 10 pg RNA [15]. GPR55 mRNA was expressed as copies per unit of 18S rRNA. Real-time qRT-PCR assays for CEA, CXCL16, CXCL17, GPR35 V2/3 and LGR5 mRNAs were described previously [7,12,13,15,16].

### 4.4. Ethical Considerations

All procedures performed in studies involving human participants were in accordance with the ethical standards of the institutional research committee and with the 1964 Helsinki Declaration and its later amendments or comparable ethical standards. Tumor samples and lymph nodes were collected after receiving patients’ written, informed consent. The study was approved by the Local Ethics Research Committee of the Medical Faculty, Umeå University, Umeå, Sweden (Registration number: 03-503, date of approval: 3 December 2003).

### 4.5. Statistical Analysis

The statistical significance of differences in mRNA levels between control lymph nodes and lymph nodes from various patient groups were analyzed using a Kruskal–Wallis one-way analysis of variance (ANOVA) test, followed by Dunn’s multiple comparison post hoc test. Statistical significance of differences in mRNA levels in primary CC tumors compared to normal colon tissues, in H&E(+) compared to H&E(−) lymph nodes and between sets of lymph nodes with different CEA levels were calculated using the two-tailed Mann–Whitney rank sum test. Correlations between different mRNA levels were analyzed using the non-parametric Spearman correlation coefficient and principal component analysis (PCA) with varimax rotation. The software utilized for statistical calculations was GraphPad Prism 8 (Graphpad Software, San Diego, CA, USA).

SPSS software (IBM Corporation, Armonk, NY, USA) was used for statistical analyses of differences between patient groups in disease-free survival time and analyses of risk for recurrent disease after surgery according to the Kaplan–Meier survival model in combination with the log-rank test and univariate Cox regression analysis. A *p*-value ≤ 0.05 was considered to be statistically significant.

## Figures and Tables

**Figure 1 ijms-23-04556-f001:**
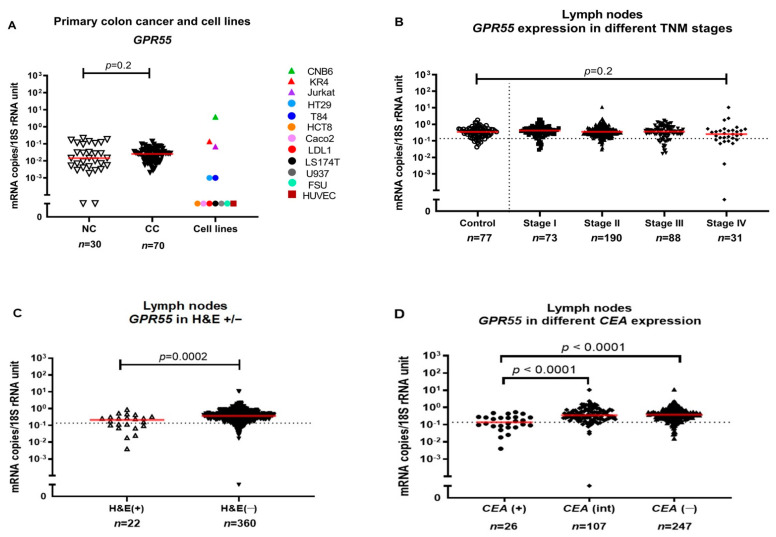
(**A**) GPR55 mRNA expression levels in resected normal colon tissues (NC), primary colon cancer tissues (CC) and in a panel of colon cancer cell lines: DLD1, LS174T, HT29, T84, HCT8 and Caco2, primary foreskin fibroblast cells (FSU), an endothelial cell line (HUVEC), a T cell line (Jurkat), two B cell lines (CNB6 and KR4) and a monocyte cell line (U937). (**B**) GPR55 mRNA expression levels in lymph nodes from non-cancerous disease patients (Control) and from colon cancer patients of different TNM stages (Stage I–IV). (**C**) GPR55 mRNA expression levels in metastatic (H&E(+)) lymph nodes and non-metastatic (H&E(−)) lymph nodes. (**D**) GPR55 mRNA expression levels in lymph nodes from CC patients divided into three groups according to their CEA mRNA levels: CEA(+) = CEA mRNA levels >3.67 copies/18S rRNA unit; CEA(int) = intermediate CEA mRNA levels, that is, 0.013–3.67 copies/18S rRNA unit; and CEA(−) = CEA mRNA levels <0.013 copies/18S rRNA unit. Red horizontal lines indicate median values. Dashed horizontal lines indicate clinical cutoff values of 0.1365 mRNA copies/18S rRNA unit for GPR55. *n* = number of lymph nodes. *p*-values were calculated by Kruskal–Wallis non-parametric ANOVA, followed by post hoc Dunn’s test for multiple comparisons in (**B**,**D**) and by two-tailed Mann–Whitney test for comparison between expression levels in (**A**,**C**).

**Figure 2 ijms-23-04556-f002:**
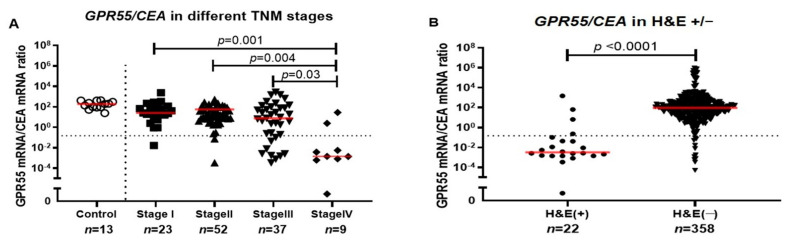
(**A**) GPR55/CEA mRNA ratio in lymph nodes from non-cancerous disease patients (control) and colon cancer patients in different TNM stages (Stage I–IV). (**B**) GPR55/CEA mRNA ratio in metastatic (H&E(+)) and non-metastatic (H&E(−)) lymph nodes. Red horizontal lines indicate median values. Dashed horizontal line shows clinical cutoff equal to 0.1481 GPR55 mRNA/CEA mRNA ratio and (*n*) number of lymph nodes. *p*-Values were calculated by Kruskal–Wallis non-parametric ANOVA, followed by post hoc Dunn’s test for multiple comparisons in (**A**) and by two-tailed Mann–Whitney test for comparison between expression levels in (**B**).

**Figure 3 ijms-23-04556-f003:**
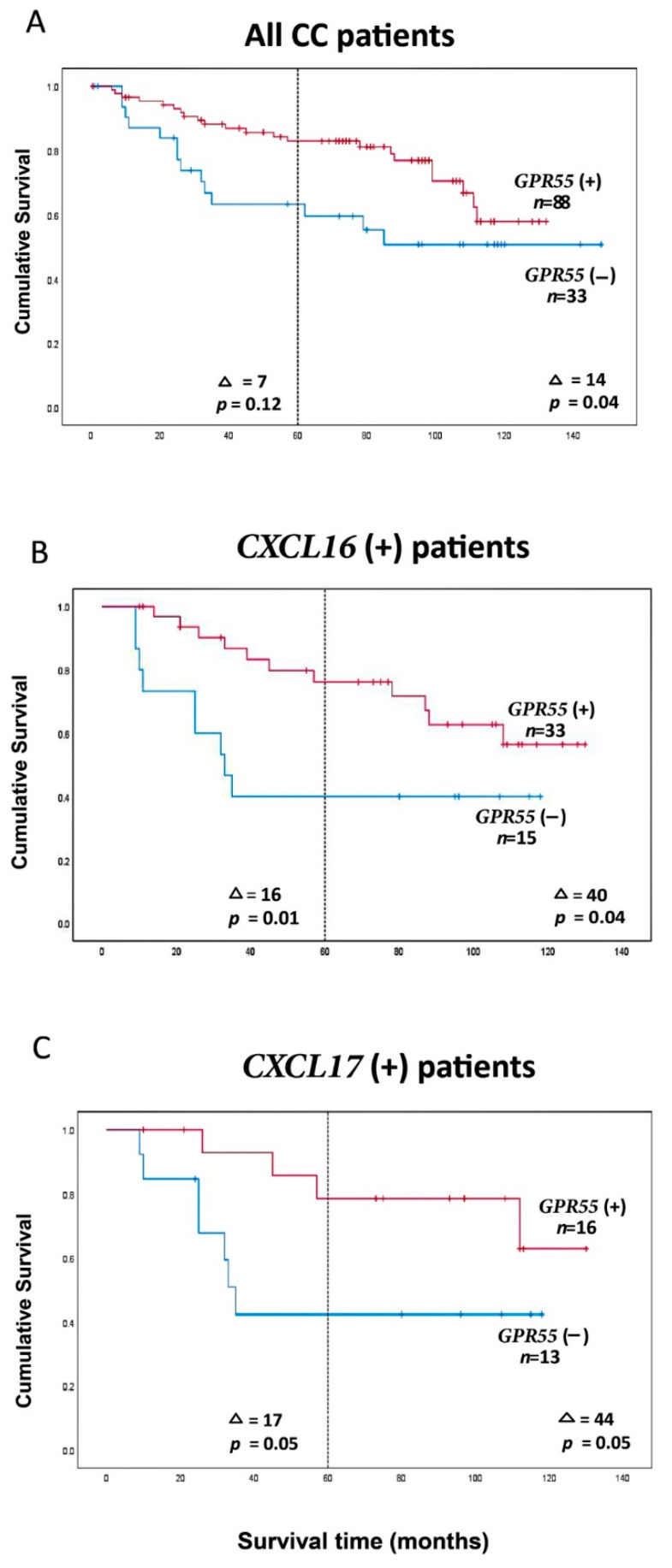
(**A**) Kaplan–Meier cumulative survival curves for all 121 CC patients. Each patient is represented by the lymph node with the lowest GPR55 mRNA value. The cutoff level between the two groups was 0.1365 GPR55 mRNA copies/18S rRNA unit. (**B**) Kaplan–Meier cumulative survival curves for GPR55(−) and GPR55(+) patients. Analysis is restricted to patients with CXCL16 mRNA levels in their highest lymph node > 7.2 mRNA copies/18S rRNA unit. The cutoff level between the two groups was 0.1365 GPR55 mRNA copies/18S rRNA unit. The number of patients was 48. (**C**) Kaplan–Meier cumulative survival curves for GPR55(−) and GPR55(+) patients. Analysis is restricted to patients with CXCL17 mRNA levels in their highest lymph node > 0.0014 mRNA copies/18S rRNA unit. The cutoff level between the two groups was 0.1365 GPR55 mRNA copies/18S rRNA unit. The number of patients was 29. Patients were followed for 12 years. Differences in disease-free survival time after surgery between the two groups are given as a ∆-value in months and statistical significance as *p*-values; *n* = number of patients in the respective group.

**Figure 4 ijms-23-04556-f004:**
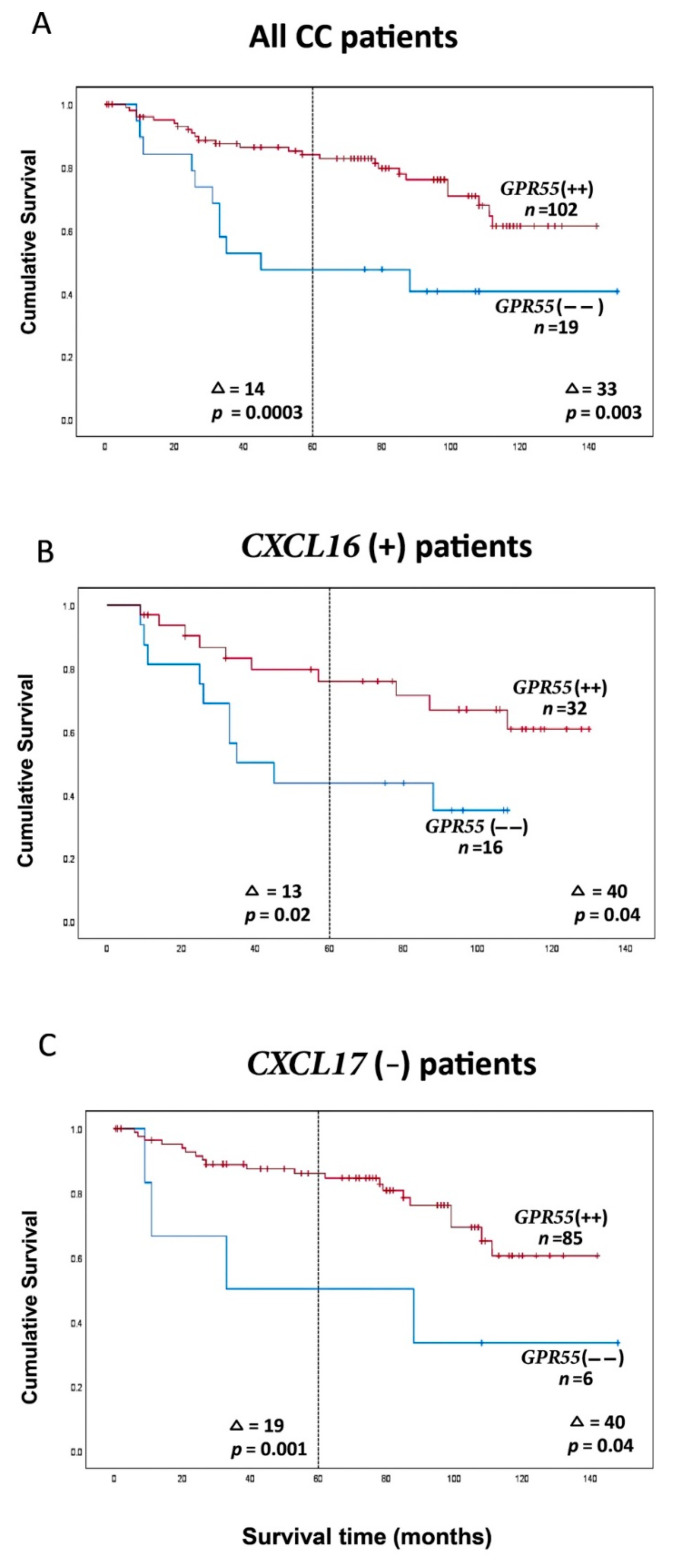
(**A**) Kaplan–Meier cumulative survival curves for GPR55(−−) and GPR55(++) patients. All 121 CC patients are included. Each patient is represented by the lymph node with the lowest GPR55 mRNA value. A GPR55 mRNA/*CEA* mRNA ratio of 0.1481 was used to divide the patients into two groups. (**B**) Kaplan–Meier cumulative survival curves for GPR55(−−) and GPR55(++) patients. The analysis is restricted to patients with CXCL16 mRNA levels in the highest lymph node > 7.2 mRNA copies/18S rRNA unit. The number of patients was 48. A GPR55 mRNA/CEA mRNA ratio of 0.1481 was used to divide the patients into two groups. (**C**) Kaplan–Meier cumulative survival curves for GPR55(−−) and GPR55(++) patients. The analysis is restricted to the CXCL17 mRNA levels in the highest lymph node < 0.0014 mRNA copies/18S rRNA unit. The number of patients was 91. A GPR55 mRNA/CEA mRNA ratio of 0.1481 was used to divide the patients into two groups. The patients were followed for 12 years. Differences in disease-free survival time after surgery between the two groups are given as a ∆-value in months and statistical significance as *p*-values; *n* = number of patients in the respective group.

**Table 1 ijms-23-04556-t001:** Correlations between *GPR55* mRNA expression levels and expression levels of other biomarkers in the lymph nodes of CC patients.

		*CEA*		*CXCL17*		*CXCL16*		*GPR35 V2/3*		*LGR5*	
		r	*p*-Value	r	*p*-Value	r	*p*-Value	r	*p*-Value	r	*p*-Value
** *GPR55* **	All CRC LNs	−0.08	0.123	−0.04	0.421	0.33	<0.0001	0.28	<0.0001	0.33	<0.0001
	TNM Stage I LNs	0.21	0.075	0.43	0.0001	0.64	<0.0001	0.58	<0.0001	0.56	<0.0001
	TNM Stage II LNs	−0.10	0.155	0.004	0.958	0.47	<0.0001	0.42	<0.0001	0.48	<0.0001
	TNM Stage III LNs	−0.18	0.098	−0.24	0.027	0.17	0.120	0.15	0.150	0.21	0.047
	TNM Stage IV LNs	−0.46	0.009	−0.38	0.036	−0.50	0.005	−0.39	0.028	−0.31	0.092

The correlation coefficients (r) and the *p*-values were calculated by two-tailed Spearman’s rank order correlation test. Levels are given as mRNA copies/18S rRNA unit.

**Table 2 ijms-23-04556-t002:** Comparative analysis of average survival time after surgery and risk for recurrence of disease in CC patients with GPR55(−) and GPR55(+) and those with GPR55(−−) and GPR55(++) lymph nodes.

Patient Group	Category	Number of Patients in Each Group Stratified by TNM Stage				Total	5-Year Follow-Up after Surgery					12-Year Follow-Up after Surgery				
							Disease-Free Survival ^a^			Risk for Recurrence ^b^		Disease-Free Survival ^a^			Risk for Recurrence ^b^	
		Stage I	Stage II	Stage III	Stage IV		Average (Months)	Difference (Months)	*p*-Value	Hazard Ratio (95% CI)	*p*-Value	Average (Months)	Difference (Months)	*p*-Value	Hazard Ratio (95% CI)	*p*-Value
*GPR55* mRNA level																
All CC patients	*GPR55*(−) ^c^	7	10	12	4	33	46	7	0.12	0.6 (0.3–1.2)	0.12	93	14	0.04	0.5 (0.3–1.0)	0.05
	*GPR55*(+)	16	42	25	5	88	53					107				
*CXCL16* ^e^	*GPR55*(−)	2	3	6	4	15	36	16	0.01	0.3 (0.1–0.8)	0.02	60	40	0.04	0.4 (0.2–1.0)	0.04
	*GPR55*(+)	8	12	10	3	33	52					100				
*CXCL17* ^f^	*GPR55*(−)	2	2	7	2	13	40	17	0.05	0.3 (0.1–1.1)	0.07	64	44	0.05	0.3 (0.1–1.1)	0.07
	*GPR55*(+)	3	5	6	2	16	57					108				
*CEA* ^g^	*GPR55*(−)	4	5	3	0	12	51	4	0.56	0.7 (0.2–2.6)	0.56	86	26	0.05	0.3 (0.1–1.1)	0.06
	*GPR55*(+)	6	17	10	0	33	55					112				
*GPR55* mRNA/CEA mRNA ratio																
All CC patients	*GPR55*(−−) ^d^	1	2	9	7	19	40	14	0.0003	0.3 (0.1–0.6)	0.001	80	33	0.003	0.4 (0.2–0.7)	0.004
	*GPR55*(++)	22	50	28	2	102	54					113				
*CXCL16* ^e^	*GPR55*(−−)	1	0	8	7	16	38	13	0.02	0.4 (0.1–0.9)	0.02	60	40	0.04	0.4 (0.2–1.0)	0.04
	*GPR55*(++)	9	15	8	0	32	51					100				
*CXCL17* ^h^	*GPR55*(−−)	0	2	1	3	6	35	19	0.001	0.2 (0.1–0.6)	0.004	73	40	0.04	0.3 (0.1–1.0)	0.05
	*GPR55*(++)	18	43	22	2	85	54					113				
*CEA* ^i^	*GPR55*(−−)	1	2	9	7	19	40	13	0.002	0.3 (0.1–0.7)	0.004	80	26	0.005	0.3 (0.1–0.8)	0.008
	*GPR55*(++)	12	28	15	2	57	53					106				

^a^ Mean survival time after surgery calculated by cumulative survival analysis according to the Kaplan–Meier model; ^b^ hazard ratio with 95% confidence interval (CI) calculated according to univariate Cox regression analysis; ^c^ CC patients divided into two groups, GPR55(−) and GPR55(+), using a cutoff of 0.1365 mRNA copies/18S rRNA unit; ^d^ CC patients divided into two groups, GPR55(−−) and GPR55(++), using a cutoff GPR55 mRNA/CEA mRNA ratio of 0.1481; ^e^ CC patient group with CXCL16 mRNA levels above 7.2 mRNA copies/18S rRNA unit; ^f^ CC patient group with CXCL17 mRNA levels above 0.0014 mRNA copies/18S rRNA unit; ^g^ CC patient group with CEA mRNA levels below 0.013 mRNA copies/18S rRNA unit; ^h^ CC patient group with CXCL17 levels below 0.0014 mRNA copies/18S rRNA unit.; ^i^ CC patient group with CEA mRNA levels above 0.013 mRNA copies/18S rRNA unit.

## Data Availability

Data used in this study provided upon request, only with permission of the authors of the original studies.

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
