# Peer review of "Prognostic Significance of GPR55 mRNA Expression in Colon Cancer"

_ijms, 2022, doi:10.3390/ijms23094556_

Round 1
Reviewer 1 Report
The authors have re-evaluated the suitability of a molecular marker for colorectal cancer using quantitative real-time PCR analysis of lymphnodes collected from surgery of CRC patients, retrospectively The marker in question is GPR55 a G-protein coupled receptor. They show, in contract to prevoius reports, that GPR55 expression in lymph-nodes is positively correlated with lack of disease recurrence and increased survival. However, this conclusion is based on ratio analysis and not analysis of GPR55 expression alone. The authors have provided good amount of statistical analsysis and persuasive argument to indicate why their findings are sound and maybe able to use in the future for designing therapy. HOwever, a mjor flaw in this manuscript they treated GPR55 as a marker not as as a functional protein for example only RNA analysis was used although they mentioned they used immunohistochemisty to localize GPR55 in tumor cells (expression poor) and in T-cell positive tumor adjacent lymph node. A short discussion on the potential function of GPR55 in context to immunesurveillance would make the paper a lot readable and relevant to the diseases prognosis. In addtion there are a some issues that authors need to explain:
1, Please show some figures of IHC to localize the expression of GPR55, if such data are availalbe as unpublished.
2. Please explain in one or two sentences in the Abstract and a paragraph in the introduction what is the biological role of GPR55 and its purported association in CRC.
3. The selection of the denominator antigen or mRNA in Figures 3 and 4 needs a little more explanation; are these ligands for CXCR55? This is important to distinguish the potential role of GPR55 as a tumor suppressor or a immune stimulator.
Author Response
Comments on the Reviewer's summary:
Thank you very much for the positive comments on our work. We think that the Reviewer perhaps has misunderstood one point, namely that our conclusions on the suitability of GPR55 mRNA as a positive prognostic marker is based solely on ratio analysis. Actually, we showed in Figure 3A that patients in the high expression group of GPR55mRNA alone [GPR55(+) group, n = 88] had a 0.5-fold decreased recurrence rate compared to the low expression group of GPR55 alone [GPR55(-) group, n = 33] at a follow-up time of 12 years (P = 0.05, respectively). The mean survival time difference between the two groups was significant (i.e., 14 months; P = 0.04). As shown in Figure 4A, the prognostic value of GPR55 mRNA was augmented by making the ratio between GPR55 mRNA and the tumor marker CEA mRNA.
We agree that analysis at the protein level would have added value to the manuscript and perhaps helped us to understand how GPR55 works in the context of colon cancer. Unfortunately, we were not able to perform for example immunohistochemistry studies because there are no good specific antibodies available.
Although our focus in this study was on the utility of GPR55 mRNA as a prognostic marker we have added a sentence on its possible role in tumor surveillance in the Abstract and at the end of the Discussion.
Answers to the Reviewer's questions:
Question 1. Please show some figures of IHC to localize the expression of GPR55, if such data are available as unpublished.
Answer to question 1: We have no reliable IHC data to include in this manuscript as explained above. Others have shown that GPR55 mRNA is expressed mainly in T-, B- and NK-cells and at very low levels in different subsets of intestinal epithelial cells. In this work we confirm that GPR55 is mainly expressed by immune cells and not by tumor cells:
1) GPR55 mRNA is highly expressed in T- and B-cell lines, and not expressed or expressed at very low levels in 4/6 CC and 2/6 CC cell lines, respectively;
2) No difference was found in the GPR55 mRNA expression levels between primary tumor and normal colon tissue;
3) GPR55 mRNA expression levels were significantly higher in H&E(-) lymph nodes compared to H&E(+) lymph nodes and in the CEA(-) and CEA (int) nodes compared to CEA(+) lymph nodes;
4) A negative correlation was found between GPR55 and the tumor marker CEA.
Question 2. Please explain in one or two sentences in the Abstract and a paragraph in the introduction what is the biological role of GPR55 and its purported association in CRC.
Answer to question 2: We have added a sentence about the possible function of GPR55 in CRC in the Abstract and in the Discussion.
Question 3. The selection of the denominator antigen or mRNA in Figures 3 and 4 needs a little more explanation; are these ligands for CXCR55? This is important to distinguish the potential role of GPR55 as a tumor suppressor or a immune stimulator.
Answer to question 3: We do not really understand this question. To the best of our knowledge there is no CXCR55 molecule. CEA, CXCL17, and CXC16 were chosen because of our previous findings that they, in contrast to GPR55, are markers for bad prognosis in CRC.
Reviewer 2 Report
An interesting paper on the prognostic value of G protein-coupled receptor 55 (GPR55) in colon cancer is submitted for review. Meyan has a few questions for the authors: 1) Have you checked if there are differences in the GPR55 level depending on the gender of the patients? 2) I am confused by the difference in the age of patients in the control group (median 23 years) and the main group (median 73 years). There are likely age differences. 3) I would like to see the results of a multivariate analysis of the levels of GPR55, CEA, CXCL17 and CXCL16 in order to understand how significant the contribution of GPR55 is for the prognosis of colon cancer.
Author Response
Question 1. Have you checked if there are differences in the GPR55 level depending on the gender of the patients?
Answer to question 1: No difference was found between the expression levels of GPR55 mRNA based on gender (P > 0.05) either in the lymph nodes or the primary tumors of CC patients. This information is now added to the Result section (page 4).
Question 2. I am confused by the difference in the age of patients in the control group (median 23 years) and the main group (median 73 years). There are likely age differences.
Answer to question 2: It is a known problem to obtain lymph nodes from age matched controls to CC patients. The control group comprises patients with inflammatory bowel disease, these patients usually get operated at a significantly younger age. However, from our data it seems that there is no or only marginal influence of age on the expression levels of GPR55 in lymph nodes since the levels were the same in control lymph nodes and nodes of CC patients in stage I and II.
Question 3. I would like to see the results of a multivariate analysis of the levels of GPR55, CEA, CXCL17 and CXCL16 in order to understand how significant the contribution of GPR55 is for the prognosis of colon cancer.
Answer to question 3: We have performed principal component analysis (PCA) with varimax rotation for the four variables GPR55, CXCL16, CXCL17 and CEA. The results extracted two principal components. The first showed 31% of variance in data and the second showed 25%. GPR55, CXCL16, and CXCL17 mRNAs variables were related to the first component and showed that CXCL16 and CXCL17 were negatively related to GPR55 (-0.52). CEA mRNA was related to the second component and showed a negative relation with GPR55 (-0.48). These data are now included in the text of the Result section and a new supplementary Figure 1 is added (shown below).
Supplementary Figure 1. Principal component analysis (PCA) was performed with varimax rotation displaying two components.

Round 2
Reviewer 2 Report
I have no more questions or comments on the article. I think that in its present form the article can be recommended for publication.